# Immunoregulatory Biomarkers of the Remission Phase in Type 1 Diabetes: miR-30d-5p Modulates PD-1 Expression and Regulatory T Cell Expansion

**DOI:** 10.3390/ncrna9020017

**Published:** 2023-02-28

**Authors:** Laia Gomez-Muñoz, David Perna-Barrull, Marta Murillo, Maria Pilar Armengol, Marta Alcalde, Marti Catala, Silvia Rodriguez-Fernandez, Sergi Sunye, Aina Valls, Jacobo Perez, Raquel Corripio, Marta Vives-Pi

**Affiliations:** 1Immunology Department, Germans Trias i Pujol Research Institute (IGTP), Autonomous University of Barcelona, 08916 Badalona, Spain; 2Pediatrics Department, Germans Trias i Pujol University Hospital (HGTiP), Autonomous University of Barcelona, 08916 Badalona, Spain; 3Translational Genomic Platform, Germans Trias i Pujol Research Institute (IGTP), Autonomous University of Barcelona, 08916 Badalona, Spain; 4Physics Department, Universitat Politècnica de Catalunya (UPC), 08034 Barcelona, Spain; 5Comparative Medicine and Bioimage Centre of Catalonia (CMCiB), Germans Trias i Pujol Research Institute (IGTP), 08916 Badalona, Spain; 6Nuffield Department of Orthopaedics, Rheumatology and Musculoskeletal Sciences (NDORMS), University of Oxford, Oxford OX1 2JD, UK; 7Pediatric Endocrinology Department, Parc Taulí Hospital Universitari, Institut d’Investigació i Innovació Parc Taulí (I3PT), Autonomous University of Barcelona, 08208 Sabadell, Spain

**Keywords:** microRNA, type 1 diabetes, biomarker, honeymoon, partial remission, childhood, immune regulation, epigenetics, islet autoimmunity, regulatory T cell

## Abstract

The partial remission (PR) phase of type 1 diabetes (T1D) is an underexplored period characterized by endogenous insulin production and downmodulated autoimmunity. To comprehend the mechanisms behind this transitory phase and develop precision medicine strategies, biomarker discovery and patient stratification are unmet needs. MicroRNAs (miRNAs) are small RNA molecules that negatively regulate gene expression and modulate several biological processes, functioning as biomarkers for many diseases. Here, we identify and validate a unique miRNA signature during PR in pediatric patients with T1D by employing small RNA sequencing and RT-qPCR. These miRNAs were mainly related to the immune system, metabolism, stress, and apoptosis pathways. The implication in autoimmunity of the most dysregulated miRNA, miR-30d-5p, was evaluated in vivo in the non-obese diabetic mouse. MiR-30d-5p inhibition resulted in increased regulatory T cell percentages in the pancreatic lymph nodes together with a higher expression of *CD200*. In the spleen, a decrease in PD-1^+^ T lymphocytes and reduced *PDCD1* expression were observed. Moreover, miR-30d-5p inhibition led to an increased islet leukocytic infiltrate and changes in both effector and memory T lymphocytes. In conclusion, the miRNA signature found during PR shows new putative biomarkers and highlights the immunomodulatory role of miR-30d-5p, elucidating the processes driving this phase.

## 1. Introduction

Type 1 diabetes (T1D) results from the immune-mediated destruction of the insulin-producing β-cells of the pancreas, which inevitably leads to the appearance of hyperglycemia. This metabolic and autoimmune disease is one of the most prevalent chronic pediatric diseases [1], and patients rely on exogenous insulin therapy for life. The β-cell attack is silent and gradual, reflected by the appearance of islet autoantibodies long before the disease is diagnosed, at which point 60–80% of these cells have already been destroyed. However, the decline in the β-cell mass is not necessarily linear; instead, patients can course with a relapsing/remitting pattern in which they retain some ability to regenerate β-cells and produce enough endogenous insulin to reduce the requirement of the exogenous one [2,3]. Clinically, this is detected only during the partial remission (PR) phase, called the “honeymoon phase”, a stage experienced by up to 80% of pediatric patients with T1D after the initiation of insulin therapy and that is characterized by low requirements of exogenous insulin and diminished glycated hemoglobin (HbA1c) levels [4]. Although the mechanisms behind this intriguing transient period have been poorly explored, it has been associated with β-cell rest, recovery, and regeneration as well as with immunological changes that could reflect an attempt to restore self-tolerance [5,6]. In fact, peripheral levels of different immune system cells and cytokines may influence the appearance and length of the PR phase [7,8,9,10].

Although metabolic biomarkers and autoantibodies have been useful to monitor and correctly diagnose patients, they do not reflect the variations of the autoimmune response or β-cell regeneration and do not predict the development of the disease. Therefore, there is an unmet need for reliable and cost-effective biomarkers for this heterogeneous disease. Many studies are currently underway to find new immunological biomarkers that can both predict disease progression and response to therapies [11] and stratify patients into different endotypes [12,13]. Furthermore, new biomarkers of remission are arising to understand the immunological mechanisms of this phase, to monitor the early course of T1D, and to stratify patients with better disease prognosis [14,15,16].

MicroRNAs (miRNAs) are a family of endogenous ∼18–25-nucleotide non-coding small RNAs that negatively regulate gene expression at the post-transcriptional level and in a sequence-specific manner. In that sense, miRNAs act on gene expression through different mechanisms, including mRNA degradation, destabilization by deadenylation, and/or translational repression [17], thus modulating many biological processes, such as immune regulation or cellular differentiation, proliferation, metabolism, and apoptosis. Since miRNAs are very stable molecules that can be found both in many mammalian cell types and in cell-free body fluids, they were recently proposed as potentially available blood biomarkers for several disorders, including autoimmunity [18,19,20]. More specifically, in T1D, some differentially expressed miRNAs (DEMs) have been linked to an aberrant T lymphocyte activation, differentiation, and function as well as β-cell apoptosis [21,22,23,24], which suggests a direct role of these molecules in regulating the onset of islet autoimmunity. Indeed, islet autoimmunity induces the expression of miR-142-3p on CD4^+^ T cells, which impairs regulatory T cell (Treg) homeostasis and functions by targeting TET2, an enzyme that maintains regions within the Foxp3 locus in a hypomethylated state, thus ensuring proper Foxp3 expression in Tregs [25]. Moreover, newly diagnosed T1D patients present a different peripheral miRNA signature when compared to control subjects or patients with long-standing disease [26,27,28,29,30,31,32], indicating that these miRNAs may reflect different stages of the disease or serve both as diagnostic and prognostic biomarkers. Because miRNAs regulate the expression of genes involved in β-cell function, inflammation, and autoimmunity [33], we hypothesize that a specific plasma miRNA profile can be identified in T1D patients during the PR phase that may reflect attempts at immunoregulation or β-cell functional improvement.

Here, we identified for the first time a unique peripheral miRNA signature in pediatric patients with T1D during the PR phase and revealed novel immunomodulatory roles for the upregulated miR-30d-5p in the context of autoimmunity in the experimental model of the disease: non-obese diabetic (NOD) mice.

## 2. Results

### 2.1. Distinct miRNA Signature during the Partial Remission Phase of Pediatric Patients with Type 1 Diabetes

To identify miRNA expression signatures during PR, small RNA was isolated and sequenced from the plasma of the discovery cohort (Table 1; clinical results from discovery cohort were previously and partially published in [14]) composed of 17 newly diagnosed children with T1D (age: 8.7 ± 3.6 years, mean ± SD), of whom 11 were remitters (age: 9.1 ± 4.3 years), and 6 were non-remitters (at eight months after diagnosis; age: 9 ± 2.8). The control group was chosen from 17 age- and sex-matched non-diabetic children (age: 8.8 ± 3.4) who visited the Germans Trias i Pujol University Hospital.

However, due to the limited amount of small RNAs of four samples at T1D diagnosis, only 13 samples out of the 17 at this time-point could be profiled on the Illumina NextSeq 1000 System. First, we focused on the comparison between remitters and non-remitters to find a specific miRNA signature that defines the PR. The hierarchical clustering heatmap revealed 16 DEMs (|log(fold change (FC))| > 1 and *p*-value ≤ 0.05); 12 of them were upregulated, and 4 were downregulated during the remission phase (Figure 1A). To identify the miRNAs with the greatest FC and the lowest *p*-value, a volcano plot was constructed, where blue dots represent downregulated miRNAs, and red dots represent the upregulated ones. Within this last group, we observed that the miR-30d-5p had the greatest FC with the highest statistical significance, followed by miR-106a-5p and miR-20b-5p (Figure 1B). Additionally, it is worth mentioning that five miRNAs (miR-106a-5p, miR-106b-5p, miR-18b-5p, miR-17-5p, and miR-20b-5p) belong to the same family of miRNAs, the MIR-17 family. Next, we performed a principal component analysis based on each sample’s miRNA expression, which could effectively differentiate the two clusters of patients; the orange are the remitters (PR), and the violet are the non-remitters (non-PR) (Figure 1C). Nevertheless, two remitter patients were clustered within the non-remitter group, which coincides with the two of the heatmap that showed a distinct profile in comparison to the other PR patients (marked with an asterisk); one of them also presented celiac disease. The biplot showing the principal component scores and the miRNAs allowed us to identify which miRNA influences one component or another. In this case, miR-25-5p, let-7b-5p, let-7c-5p, and miR-10393-3p had the most influence in determining the non-PR group (Appendix A). To examine which of these DEMs during PR were present in other comparisons, we made Venn diagrams. We found that the miR-30d-5p was also differentially expressed between the PR phase and both the control group (FC = 1.777, *p*-value = 0.019; Table 2) and T1D diagnosis (FC = 1.780, *p*-value = 0.053), which gives it more potency as a specific biomarker of remission. Other miRNAs were more related to the general disease status, such as miR-17-5p, which was further differentially expressed between the control group and the diagnosis of T1D in those future remitter patients (Figure 1D). Table 2 shows the DEMs (|log(FC)| > 1 and *p*-value ≤ 0.05) with validated target genes between PR vs. non-PR (#14) and PR vs. their diagnosis time-point (#20) as well as the ones that were also differentially expressed between PR vs. controls (#12). The complete list of DEMs between PR vs. controls (#62) and T1D diagnosis vs. controls (#16) can be found in Appendix A.

### 2.2. Validation of Differentially Expressed miRNAs during the Partial Remission Phase

Then, we selected some of the miRNAs (miR-142-3p, miR-17-5p, miR-106b-5p, miR-20b-5p, let-7b-5p, and let-7c-5p) with the highest number of normalized counts per million and that were present in at least 70% of the forty-seven profiled samples from the discovery cohort in order to validate their expression using single-assay RT-qPCR in another cohort of patients, the validation cohort (Table 1). This cohort was composed of 15 age- and sex-matched non-diabetic controls (age: 9.7 ± 3.6), 8 newly diagnosed children with T1D (age: 11.6 ± 2.7), 10 remitter patients (age: 11.8 ± 2.9), and 9 non-remitter patients (age: 7.3 ± 3.7). Five out of six miRNAs were validated as differentially expressed; miR-142-3p was slightly upregulated, and both miR-17-5p and miR-106b-5p were significantly upregulated during PR in comparison to the non-PR group (Figure 2A–C). Both let-7b-5p and let-7c-5p were also slightly downregulated at this stage (Figure 2E,F). No differences were found regarding the levels of miR-20b-5p between the PR and non-PR groups (Figure 2D).

### 2.3. Potential Enriched Biological Functions and Pathways under the Regulation of Differentially Expressed miRNAs during the Partial Remission Phase

To identify potential biological functions and pathways affected by the miRNA signature during PR, we performed in silico miRNA functional analyses based on the inferred miRNA target genes using the DIANA-miRPath v3.0 web server. In this case, we only used miRNAs with experimentally validated target genes, thus ruling miR-10393-3p and miR-10395-3p out of the 16 DEMs from the analysis. Biological processes and pathways were investigated using the DIANA-TarBase v7.0 and a Fisher’s exact test, and the false discovery rate (FDR) method (*q*-values) was used to calculate the enriched targeted ones. The interactive graph of similar non-redundant gene ontology (GO) terms was retrieved from the web server REVIGO, and it clearly showed two different clusters of enriched biological processes: one that comprises different metabolic processes (e.g., mRNA metabolic process, cellular lipid metabolic process, or cellular protein metabolic process) (upper cluster) and one comprising different signaling pathways related to the immune response (e.g., toll-like receptor TLR6:TLR2 signaling pathway or Fc-gamma receptor signaling pathway involved in phagocytosis), apoptosis (e.g., intrinsic apoptotic signaling pathway), or stress (e.g., stress-activated MAPK cascade) (lower cluster). These are relevant biological functions under the pathogenesis of T1D (Figure 3).

The pathway analysis was performed using Kyoto Encyclopedia of Genes and Genomes (KEGG) annotations. Clustering the miRNAs based on their influence on molecular pathways (Figure 4, top) indicated that miR-17-5p, miR-106b-5p, miR-20b-5p, and miR-106a-5p—all from the MIR-17 family—appeared to target similar pathways and were the ones with a broad effect among all the enriched miRNAs. Of those, other interesting pathways were Wnt, mTOR, FoxO, and MAPK signaling pathways. The 10 most enriched KEGG pathways are listed in the table (Figure 4, bottom), with fatty acid biosynthesis being the most statistically significant. Interestingly, the TGF-β signaling pathway was among the top 10, which is known for its crucial role in immune homeostasis and tolerance.

### 2.4. In Vivo Inhibition of miR-30d-5p Potentiates Treg Expansion by Increasing CD200 Levels

MiR-30d-5p was the most differentially expressed miRNA during the PR phase in comparison to the non-remission, and its role in the immune system or T1D has been poorly explored. This miRNA was enriched for biological processes such as the immune system, response to stress, cell death, or insulin receptor signaling. In addition, miR-30d-5p was among the miRNAs that presented the highest number of immune-related target genes, including some that have a direct role in the activation of T lymphocytes, such as CTLA-4 or CD226 (Appendix A). Thus, to address the immunological relevance of miR-30d-5p in vivo, we next analyzed the effect of miR-30d-5p inhibition in nine-week-old prediabetic NOD mice. We employed a well-known Locked Nucleic Acid (LNA)-miRNA-inhibitor that, when administered systematically, has been shown to accumulate in different tissues, including lymphoid tissues and the pancreas, thus promoting in vivo miRNA silencing [25]. The miR-30d-5p inhibitor or control inhibitor (negative control A probe; miRNA control group) were applied four times by intraperitoneal (i.p.) injection at 9 mg/kg every three days, whereas in the sham group, mice were treated with 200 µL PBS following the same pattern. Along with the treatment with the miRNA inhibitor, mice exhibited normal glycosuria and glycemia at the end of the study and maintained weight within a normal range, which led us to think that the inhibition of this miRNA did not have a toxic effect on mice, nor did it significantly advance the onset of diabetes. Moreover, the viability of splenocytes and pancreatic lymph node (PLN) cells was optimal in the three groups, despite small significant differences (Appendix A).

Figure 5A shows representative fluorescence-activated cell sorting (FACS) plots indicating the percentage of CD4^+^CD25^+^FoxP3^+^ Tregs in PLN of the sham (left, grey), control inhibitor (middle, blue), and miRNA inhibitor (right, red) treatment groups. At the end of the study, the blockade of miR-30d-5p resulted in an increased percentage of Treg cells in the PLN in comparison to both the sham group and the miRNA control group (Figure 5B). We then classified the different predicted and validated target genes for miR-30d-5p (approx. 2.000 targets) into three groups, (1) regeneration, (2) metabolism, and (3) immune system (Appendix A). Regarding the immune system group, we found interesting target genes such as *CD200*, which is associated with Treg expansion [34]. To demonstrate the efficacy of the delivered miR-30d-5p inhibitor in the pancreatic tissue and local immune cells, we analyzed the expression of *CD200* in the remaining immune cells from PLN. We observed higher levels of *CD200* mRNA in the absence of miR-30d-5p in comparison to the miRNA control group, validating the in vivo effect of miR-30d-5p inhibition (Figure 5C). Interestingly, we found a significant positive correlation between *CD200* expression levels and the percentage of Treg cells, further confirming the role of this *CD200* inhibitory immune checkpoint in immune tolerance and regulation (Figure 5D).

### 2.5. In Vivo Inhibition of miR-30d-5p Reduces PD-1 Expression on Splenocytes

Figure 6A,C shows representative FACS plots indicating the percentage of CD4^+^PD-1^+^ T cells and PD-1^+^ Tregs, respectively, in the spleen of the sham (left, grey), control inhibitor (middle, blue), and miRNA inhibitor (right, red) treatment groups. At the end of the study, the blockade of miR-30d-5p resulted in a significant reduction in the percentage and also total cell numbers of CD4^+^ T cells positive for PD-1 in comparison to both the sham group and the miRNA control group (Figure 6B). Furthermore, this PD-1 expression decline was very pronounced in Treg cells, both in percentage and total cell number (Figure 6D). *PDCD1* (PD-1 gene) is not a direct target gene of miR-30d-5p, but we found *PRDM1*, a repressor of PD-1, which is [35] (Appendix A). Thus, in order to link the low expression of PD-1 with the upregulation of one of its repressors in the absence of miR-30d-5p, we analyzed the expression levels of both *PRDM1* and *PDCD1* on the splenocytes. Although we could not find differences regarding the expression of *PRDM1* between miRNA inhibitor and control groups (Figure 6E), we did find decreased *PDCD1* mRNA levels, confirming the results obtained by flow cytometry (Figure 6F).

We further checked the expression levels of *TGFBR2* and *PRDM1* in the remaining PLN and of *TGFBR2* and *CD200* in the splenocytes, but no differences were found between the miRNA control and miRNA inhibitor groups (Appendix A).

### 2.6. In Vivo Inhibition of miR-30d-5p Displays Changes in Additional T Cell Differentiation Subsets

To examine the effect of the miR-30d-5p blockade on the maturation of the immune system and different immune cell populations, we analyzed CD4^+^ and CD8^+^ T lymphocytes (differentiating between naïve, central memory (CM), effector memory (EM), and pre-effector-like T cells), B lymphocytes, and conventional dendritic cells (DCs). First, no differences in the percentages of total CD4^+^ and CD8^+^ T cells, DCs, or B lymphocytes were found between the three treatment groups in either PLN (Appendix A) or the spleen (Appendix A). Representative FACS plots for the analysis of B lymphocytes and DCs in PLN and the spleen are respectively shown in Appendix A.

Figure 7A,C show representative FACS plots indicating the percentage of naïve, CM, EM, and pre-effector-like CD4^+^ and CD8^+^ T cells, respectively, in the PLN of miRNA control and miRNA inhibitor groups. Although not significant, we found that the miR-30d-5p blockade led to an increase in the percentage of CD4^+^ EM T lymphocytes (Figure 7B) together with a decrease in the percentage of CD8^+^ pre-effector-like T lymphocytes (Figure 7D) and an increase in the CD8^+^ CM ones (Figure 7E).

Regarding the spleen, representative FACS plots indicating the percentage of naïve, CM, EM, and pre-effector-like CD4^+^ T lymphocytes can be found in Figure 8A. In this organ, the miR-30d-5p blockade tended to decrease the percentages and total cell numbers of CD4^+^ pre-effector-like T cells (Figure 8B) and significantly reduced the percentages of the CD4^+^ CM T cells, which could also be observed regarding their numbers (Figure 8C). No differences were found within splenic CD8^+^ T cell subsets.

### 2.7. In Vivo Inhibition of miR-30d-5p Tends to Increase Leukocyte Islet Infiltration

The insulitis score was determined at the end of the short treatment. As expected, mice in the sham group showed a similar insulitis degree (0.96 ± 0.39, mean ± SD) as the miRNA control group (1.182 ± 0.25) (Figure 9A). Mice treated with the miR-30d-5p inhibitor showed a biological trend toward an increased insulitis score (1.33 ± 0.51) in comparison to both control groups (Figure 9A); 37.3% of their islets were infiltrated or destroyed (scores from two to four), whereas only 27.3% and 33.5% of the islets were destroyed in the sham and the miRNA control groups, respectively (Figure 9B).

## 3. Discussion

In recent years, much progress has been made in the study of the different factors that affect the progression of T1D and its immunopathology, revealing that this disease is more heterogeneous than initially thought [36]. In fact, there is great inter-subject variability in terms of age at diagnosis, autoantibodies, genetics, metabolic control, rate of progression, and immune activity, which has led to the hypothesis that there are different T1D endotypes [37]. This makes the search for new biomarkers essential to understand the different courses of the disease and stratify patients. In this regard, although up to almost 80% of patients experience the PR phase after the initiation of insulin therapy, it is still an under-explored period, although it is of great interest both metabolically and immunologically.

Circulating microRNAs have huge potential as a novel class of non-invasive biomarkers that reflect disease activity and attempts at β-cell regeneration or immune regulation [22,38]. Here, we identified for the first time a circulating miRNA signature for PR in a longitudinal pediatric cohort of patients and described an immunoregulatory role for miR-30d-5p in the NOD mouse model by modulating miRNA expression.

The onset of islet autoimmunity has been associated with several dysregulated miRNAs both in circulation [28,32,39,40,41,42] and in peripheral blood mononuclear cells or T lymphocytes [25,31,43,44]; however, only a few studies have addressed changes in miRNA levels over time upon diagnosis [26,27]. Specifically in children, one of the first studies found that miR-25 levels in sera are associated with improved glycemic control and stimulated C-peptide three months after diagnosis [29]. Similarly, levels of the miR-23~24~27 cluster in newly diagnosed children predict C-peptide loss over time and are upmodulated upon disease progression [30]. A study conducted on the Danish Remission Phase Cohort found that the miR-197-3p at three months—when PR usually occurs—was the strongest predictor of residual β-cell function one year after diagnosis in children with T1D [45]. Nevertheless, none of them evaluated specific changes in miRNA levels during the PR phase, so that was our first aim. On the one hand, we identified 12 upregulated miRNAs in pediatric remitter patients versus non-remitters. Of those, five belonged to the MIR-17 family (miR-20b-5p, miR-17-5p, miR-106a-5p, miR-106b, and miR-18b-5p). Some of these miRNAs have been previously associated with β-cell apoptosis/regeneration processes and immunomodulation. For instance, the inflammatory microenvironment led β-cells to downregulate the expression of miR-17, a miRNA that negatively affects ERAP1 mRNA, impairing the processing of preproinsulin signal peptide antigen and limiting its recognition by autoreactive CD8^+^ T cells [46]. Thus, the upregulation of miR-17 during the PR phase could be related to the decreased inflammatory environment and might be involved in decreasing β-cell visibility to the immune system. Moreover, whereas miR-17-92 and miR-106b-25 clusters positively regulate β-cell proliferation and insulin secretion in mice and are important for normal endocrine function [47,48], miR-20b can inhibit T cell proliferation and activation by targeting NFAT [49]. Hence, both facts would contribute to the hypothesis that behind PR, there are processes of immunoregulation and β-cell regeneration that are controlled by epigenetics. On the other hand, we identified four downregulated miRNAs during PR in comparison to non-PR, including let-7b-5p and let-7c-5p. Recently, it was reported that let-7b-5p overexpression impairs insulin production and secretion and inhibits β-cell proliferation in mice [50,51], which could be related to the diminished residual β-function in patients without remission. Notably, of the 16 DEMs, miR-10395-3p, miR-10393-3p, and miR-1277-3p are described for the first time in relation to T1D. Furthermore, the GO analysis revealed several enriched metabolic, immunological, apoptosis, and stress processes, which are pathways closely related to T1D immunopathogenesis. The functional annotation of genes regulated by these miRNAs also implies that TGF-β and FoxO signaling pathways (among others) may be involved in the occurrence of PR, which can control the development and function of Foxp3^+^ Tregs [52].

MiR-30d-5p was the most upregulated miRNA during PR, where insulin synthesis and secretion are improved. This is a glucose-regulated miRNA that has been associated with both the induction of insulin production by activating MafA in pancreatic β-cells and the protection of β-cell function from impairment caused by proinflammatory cytokines [53,54]. Even though we are uncertain about which cells are producing this miRNA and thus contributing to its circulating pool, it is known to be highly expressed by pancreatic β-cells, suggesting that they are likely one of its primary sources [54]. In cancer cells, miR-30d-5p induces IL-10 expression (an immunosuppressive cytokine), at least in part by repressing the *GALNT7* gene, resulting in pro-metastatic effects in vivo [55]. Regarding autoimmunity, miR-30d-5p is capable of regulating the microbiome in the experimental autoimmune encephalomyelitis model, which in turn results in enhanced immunosuppression and the amelioration of the multiple-sclerosis-like symptoms [56]. In the context of T1D, miR-30d is found both up- [57] and downregulated [32] in the plasma of people with T1D versus control subjects and is also increased in exosomes from lactating mothers with T1D [58]. Nevertheless, little is known about its immunological functions.

In this work, multiple lines of evidence point to a link between miR-30d-5p and immunoregulatory processes. First, by analyzing its potential targets, we found that miR-30d-5p is mainly enriched for fatty acid biosynthesis pathways, which have key roles in T cell development and immune responses [59]. Second, we managed to classify some of its predicted and validated targets and discovered that this miRNA affects genes such as *CD200*, *CCL5*, *CTLA4*, *NFAT5*, *PRDM1*, and *TGFBR2*. In addition, the direct inhibition of miR-30d-5p in the NOD mouse model led to changes in different immune cell subsets in secondary lymphoid organs and immune cell infiltration in the pancreatic islets.

Upon miR-30d-5p inhibition, the levels of Tregs were significantly increased in PLN. We found that this expansion could be explained by the increase in the expression levels of *CD200* in PLN cells, which is a direct target of miR-30d-5p [55]. In fact, CD200-CD200R-mediated immunosuppression can occur through the induction of FoxP3^+^ Tregs [60,61]. Since miR-30d-5p is upregulated in PR vs. non-PR, we should expect lower levels of Tregs along this phase. In a longitudinal study that included the same pediatric patients, we previously found decreased levels of peripheral Tregs at the PR phase in comparison to non-remission, even after 12 months from T1D diagnosis [14], which is accordant with other studies showing diminished Treg levels during the honeymoon phase or after one year [7,62]. Furthermore, although we could not confirm a miR-30d-5p/Blimp-1(*PRDM1*)/PD-1(*PDCD1*) axis acting on splenocytes of NOD mice, miR-30d-5p is probably influencing PD-1 expression through other mechanisms that need further research. Interestingly, a recent longitudinal study found a relationship between the PR phase and the restoration of the PD-1/PD-L1 axis on peripheral T cells, suggesting an immunoregulatory mechanism that is absent in patients without remission [15]. These results follow what we saw in the spleen since a higher expression of miR-30d-5p (as happens during PR) was associated with an increased expression of PD-1. Moreover, the reduced levels of pre-effector-like T lymphocytes together with the increased levels of EM and CM T lymphocytes upon miR-30d-5p inhibition could reflect the wave of differentiation into the effector and memory phenotypes, which have a key role in amplifying inflammation. In fact, the expansion of CM T cells might boost the pathogenic potential of the peripheral T cell pool and favor autoimmunity. Finally, the slight increase in the immune cell infiltration into the pancreatic islets could be related to the enhanced effector function of these T lymphocytes. Therefore, the upregulation of miR-30d-5p during the PR might be related to a decreased immune cell infiltration in the islets of Langerhans and the amelioration of the inflammatory microenvironment, with the consequent prevention of β-cell apoptosis. Nevertheless, it would be interesting to identify the nature of the infiltrating T lymphocytes. We must take into consideration that the observed insulitis might be enriched on Tregs since there is an increase in these regulatory cells in the draining lymph nodes. Taking all these data together, we hypothesize that in the absence of miR-30d-5p (non-remission scenario), T lymphocytes do not receive the inhibitory signal through PD-1 because of its low expression levels, which could potentiate their effector functions and their contribution to the inflammatory immune cell infiltrate into the pancreatic islets. At the same time, increased levels of Tregs are generated—in part due to the *CD200* upregulation—to try to tackle this enhanced immune response.

This study has limitations that must be taken into account when interpreting the results. First of all, the sample size is relatively small, yet we were able to validate some of the miRNAs by RT-qPCR, a highly recommended step [63]. Even though we verified the absence of batch effects and the suitability of the normalized data for the differential expression analysis, another limitation in the RNA sequencing experiment was the absence of technical replicates between runs. Furthermore, although the NOD model spontaneously recapitulates autoimmunity in pancreatic islets, there are key differences in disease development between NOD mice and humans. Another important point to consider is that despite having been able to indirectly test the effect of the miR-30d-5p inhibitor by analyzing the up- or downregulation of some miRNA target genes, it could be interesting to test its specific delivery to T lymphocytes. In this sense, numerous studies have used inhibitory/mimic miRNAs coupled to fluorescent molecules to see their accumulation in vivo in different tissues, including lymph nodes [64,65,66,67].

## 4. Materials and Methods

### 4.1. Human Sample Collection and T1D Remission Follow-Up

The longitudinal discovery cohort for the small RNA sequencing was composed of 17 pediatric patients with new-onset T1D (11 remitters and 6 non-remitters) and 17 age- and sex-matched non-diabetic control subjects. The validation cohort was composed of non-paired samples that included 8 pediatric patients with new-onset T1D, 10 remitters, 9 non-remitters (5 of them also included in the discovery cohort), and 15 age- and sex-matched control subjects. All patients fulfilled the American Diabetes Association classification criteria for T1D [68], with at least one positive anti-islet autoantibody at disease onset (to insulinoma-antigen 2 or glutamic acid decarboxylase). Inclusion criteria were 4–18 years of age and normal body mass index according to the Spanish Body Mass Index pediatric cohort growth chart [69]. Exclusion criteria were being under immunosuppressive or anti-inflammatory treatment, type 2 diabetes, pregnancy, compromised kidney function, or liver diseases. The same inclusion/exclusion criteria were used for non-diabetic controls.

T1D data collection occurred for over a year in two University Hospitals of our geographical area, Germans Trias i Pujol (Badalona) and Parc Taulí (Sabadell). Blood samples of 6 mL were obtained at disease onset and PR or 8 months for non-remitter patients in EDTA tubes (BD Biosciences, San Jose, CA, USA). Blood samples from control subjects without T1D were also acquired following the same protocol. Plasma samples were always obtained within the first hour after venipuncture. The tube containing the blood sample was centrifuged at 1900× *g* at 4 °C for 10 min. The plasma was then aspirated to a 1.5 mL Eppendorf (without disturbing the intermediate layer containing white blood cells and platelets) and centrifuged at 16,000× *g* at 4 °C for another 10 min to remove additional cellular nucleic acids bound to cellular debris. Finally, 250 μL of clear plasma was pipetted into a 1.5 mL Eppendorf and stored at −80 °C.

At disease onset, all samples were collected between 1 and 14 days after diagnosis. To measure PR, we calculated the insulin dose-adjusted HbA1c (IDAA1c) using both the HbA1c value and the insulin requirement as HbA1c (%) + [4 × insulin dose (U/kg/day)]. An IDAA1c equal to or lower than nine indicated the PR phase [70]. Given that the highest percentage of patients in PR is detected within the first 2–6 months after diagnosis, those who did not meet the criteria of PR after 8 months were defined as non-remitters.

### 4.2. Clinical and Laboratory Testing

Clinical descriptors on each patient and control subject were collected, including age, sex, and body mass index. Blood samples from patients with T1D were obtained for centralized measurement of HbA1c, basal non-fasting C-peptide (which reflects residual insulin storage), genetics, and immunology; insulin requirements were also recorded. HbA1c was determined by high-performance liquid chromatography (ADAMS A1c HA-8180V, Arkray, MN, USA) in all patients at each time point. Basal non-fasting C-peptide was determined by ELISA (Architect i2000, Abbott, IL, USA) in both controls and patients at each time point.

### 4.3. Total RNA Isolation

Total cell-free and exosomal RNA, including miRNA, was isolated from 200 µL of plasma from the discovery cohort using the miRNeasy Serum/Plasma Advanced Kit (Qiagen, Hilden, Germany), according to the manufacturer’s instructions. RNA purity, integrity and concentration were determined using TapeStation 2200 (Agilent High Sensitivity Screen Tape, Agilent Technologies Inc., Santa Clara, CA, USA). RNA was stored at −80 °C until use.

### 4.4. RNA Library Preparation, Sequencing, and Data Analysis

After the isolation of RNA, 1 µg was used to prepare RNA libraries by D-Plex Small RNA-seq Kit (Diagenode, Liege, Belgium), following the manufacturer’s instructions. After PCR amplification, size selection of fragments and adapter dimer removal were conducted in a 6% polyacrylamide gel (Invitrogen, Carlsbad, CA, USA), and library quality controls were assessed with a TapeStation 2200 using a High Sensitivity D1000 Screen Tape (Agilent). Then, small RNA libraries were sequenced on the Illumina NextSeq 1000 System (SBS-based sequencing technology, Illumina, San Diego, CA, USA) in a run of 92 and 2 × 71 cycles and a high output sequencing mode. Data were retrieved from the sequencer in the form of fastq files. The fastq files of the same sample corresponded to different runs of the same library. In this study, up to six runs were performed for each sample to achieve the desired sequencing depth (ranging from 1.6 to 35.1 million reads depending on the sample). Samples were randomly distributed among the six sequencing runs regardless of the group to which they belong.

Trimming steps were further conducted using the Cutadapt tool. This trimming included the removal of the first 16 bp of each read (corresponding to unique molecular identifiers), the polyA tail, and Illumina adapter sequences. Additionally, trimmed sequences of less than 18 bp in length were discarded. After trimming, the quality of the reads (Fastq files) was assessed with FastQC. All the reads were treated as single-end reads, a fact that allowed merging the reads from different runs according to their sample of origin with the multiQC tool. Next, the Subread/Rsubread package was used to map the sequencing reads to the genome of reference and quantify the aligned reads. For the read summarization/quantification step, annotations for precursor and mature miRNAs were obtained from the miRBase v22 database. First, mapped reads were quantified using the mature miRNA annotations contained in miRBase only. Then, unassigned reads were further quantified using the remaining small RNA annotations from miRBase (for pre-miRNA) and Gencode (for other small ncRNAs) databases. Filtering and normalization steps were performed using edgeR 3.34.1 and Limma v.3.48.3 packages from Bioconductor in R. Here, a minimal rule was applied to keep only transcripts that had at least one count per million in at least five samples, and the trimmed mean of M-values normalization method was performed to eliminate composition biases between libraries. Different types of quality controls were also performed (multi-dimensional scaling plot analysis, Euclidean distances between samples) to check that the normalized data were appropriate for the differential expression analysis; by doing so, no outliers or batch effects were detected.

For differential expression testing, the Limma’s package v.3.48.3 was used, specifically the limma-voom pipeline. Normalized data were transformed to log_2_, and DEMs were selected by adjusting a linear model with empirical Bayes moderation of the variance and, in the case of paired samples, a paired design was considered. Data were adjusted for multiple testing to obtain strong control over the FDR using the Benjamini and Hochberg method; however, since these criteria yield too few small RNAs, unadjusted *p*-values of ≤0.05 were considered for the significance criteria. Therefore, miRNAs with a *p*-value ≤ 0.05 and log_2_(FC) >1 were considered upregulated, whereas those with log_2_ < 1 were considered downregulated. The data for this study were deposited in the European Nucleotide Archive at EMBL-EBI under accession number PRJEB58187 (https://www.ebi.ac.uk/ena/browser/view/PRJEB58187, accessed on 22 December 2022).

### 4.5. Gene Targets for miRNAs

In this study, the multiMiR Bioconductor’s package was used to identify miRNA target sites in different databases (miRecords, miRTarBase, and TarBase for validated targets; DIANA-microT, ElMMo, MicroCosm, miRanda, miRDB, PicTar, PITA, and TargetScan for predicted targets). In order to classify the vast number of validated gene targets for each miRNA, a list of keywords was generated and distributed in three different groups: (1) metabolism, (2) regeneration, and (3) immune system. In this way, the metabolism group was composed of keywords like “mTOR signaling” or “insulin signaling pathway”, while the immune system group was composed of words such as “T cell activation” or “dendritic cell”. Then, using a computer logarithm, the *Entrez summary* of each gene was searched for those keywords, and the genes were consequently classified into one group or another.

### 4.6. Gene Ontology and Pathway Analysis

miRNA GO and pathway analysis were performed using the open-access web server DIANA-miRPath v3.0 (http://www.microrna.gr/miRPathv3, accessed on 13 September 2022) [71] using the 14 DEMs with validated target genes between PR and non-PR groups to search for potential biological pathways under their regulation. Biological processes and enriched pathways were investigated using the DIANA-TarBase v7.0 [72], a database that provides high-quality, manually curated and experimentally validated miRNA–target interactions. Significance levels were calculated by using Fisher’s exact test meta-analysis method with Benjamini–Hochberg’s FDR correction (*q*-value < 0.05). The statistically significant biological processes and their corresponding *q*-values were introduced in the web server REVIGO (http://revigo.irb.hr/) [73], which takes long lists of GO terms and summarizes them by removing the redundant ones. Interactive graphs showing the relationship between biological processes link highly similar GO terms by edges (using the SimRel semantic similarity measure), where the line width indicates the degree of similarity, and the color of the bubbles is the user-provided *p*-value.

Functional enrichment analysis of miRNA target genes was performed using pathway annotation from the KEGG database and a posteriori of the statistical analysis. In this mode, the server identifies all the significantly targeted pathways by the selected miRNAs. The enrichment analysis is first carried out by the server, and the significance levels (*p*-values) between each miRNA and each pathway are computed. Subsequently, for each pathway a merged *p*-value is extracted by combining the previously calculated significance levels using the Fisher’s exact test meta-analysis method and Benjamini–Hochberg’s FDR (*q*-value) to compensate for multiple testing. Since comparable miRNAs are clustered together, the heatmap makes it possible to identify miRNA subclasses or pathways that define them.

### 4.7. Quantitative RT-qPCR

To validate the small RNA-seq results, RT-qPCR was performed on human plasma samples from the validation cohort. Because plasma samples hemolyzed during acquisition can be contaminated by erythrocyte miRNAs [74], the degree of hemolysis was determined based on the optical density at 414 nm (absorbance peak of free hemoglobin) by spectrophotometry (Nanodrop 1000 Spectrophotometer, ThermoFisher Scientific, Waltham, Massachusetts, USA), and the severely hemolyzed samples (OD_414_ > 0.3) were discarded (Appendix A). Then, RNA isolation was conducted as described above. RNA was reverse transcribed to cDNA with the TaqMan™ Advanced miRNA cDNA Synthesis Kit (ThermoFisher Scientific) following the manufacturer’s instructions and by using the Veriti^®^ Thermal Cycler (ThermoFisher Scientific). To monitor retrotranscription reproducibility, we spiked in 5′-phosphorylated *Arabidopsis thaliana* miR-159a (ath-miR-159a, uuuggauugaagggagcucua), a synthetic oligonucleotide for exogenous control (ThermoFisher Scientific), to cDNA synthesis. Briefly, poly(A) polymerase was used to add a 3′-adenosine tail to the miRNA, which underwent adaptor ligation at the 5′ end. Then, a Universal RT primer (which also incorporates an adaptor) bound to the 3′ poly(A) tail and the miRNA was reverse transcribed. To improve the detection of low-expressing miRNA targets, the cDNA was next pre-amplified using the Universal forward and reverse miR-Amp Primers (which bind to the adaptors) and miR-Amp Master Mix (ThermoFisher Scientific). The 50 µL miR-Amp reaction product was stored at −20 °C until use. Amplified cDNA was 1:10 diluted, and miRNAs were profiled by RT-qPCR using the TaqMan™ Fast Advanced Master Mix (Applied Biosystems, Waltham, MA, USA) with TaqMan Advanced miRNA Assays (ThermoFisher Scientific) in 15 µL PCR reactions in triplicate. Table 3 shows the list of the TaqMan Assays used. MiRNAs to validate were chosen based on (1) the number of reads obtained in the small RNA-seq and their wide expression in most of the samples (>70%), (2) miRNAs most differentially expressed between PR and non-PR patients, and (3) target genes involved in immune system functions. Plates were run on a LightCycler^®^480 RT-PCR machine (Roche, Mannheim, Germany).

All analyzed miRNAs showed a C_t_ < 30. Relative values were calculated with the 2^−∆Ct^ method [75], and results are given as arbitrary units. Currently, a universally accepted normalization strategy based on endogenous miRNAs is still lacking. NormFinder, an algorithm for identifying the optimal normalization gene among a set of candidates [76], was used to identify the most stable miRNA within our normalized small-RNA seq data to be used as an endogenous control. In our case, the miRNA with the best stability value was miR-16-1-3p. Raw C_t_ of miR-16-1-3p in samples from the validation cohort can be found in Appendix A.

### 4.8. Mice

Wild-type NOD mice were originally obtained from the Jackson Laboratory (Bar Harbor, ME, USA) and then kept in the Animal Facility of the Centre de Medicina Comparativa i Bioimatge de Catalunya (CMCiB) under specific pathogen-free conditions. The colony was subjected to a 12 h dark/12 h light cycle and controlled temperatures between 19–23 °C with 40–60% humidity and fed with ad libitum access to acidic water at pH 5 and irradiated Teklad Global 18% Protein Rodent Diet (Harlan, Indianapolis, IN, USA). In this study, only prediabetic NOD females of 9 weeks of age were used. In order to detect and exclude mice with T1D, glycosuria levels were monitored weekly using urine test strips (Combiscreen, Analyticon Biotechnologies AG, Lichtenfels, Germany), and T1D was confirmed when glycemia rose above 300 mg/dL in one glucotest control (OneTouch Verio Reflect^®^, LifeScan IP Holdings, LLC., Zug, Switzerland).

### 4.9. In Vivo miR-30d-5p Inhibitor Administration

The mature nucleotide sequence of mmu-miR-30d-5p (5′-UGUAAACAUCCCCGACUGGAAG-3′) was obtained from www.mirbase.org (accessed on 3 May 2022), which is homologous between mice and man. Here, we used an antisense oligonucleotide, called miRCURY LNA^TM^ miRNA inhibitor, for miR-30d-5p (LNA-anti-miR-30d-5p) and a miRNA inhibitor negative control A (scrambled LNA) (Exiqon Co., Copenhagen, Denmark). Thus, three groups composed of six prediabetic NOD mice of 9 weeks of age each were respectively treated with (1) miRNA-inhibitor (inhibitor probe mmu-miR-30d-5p), (2) miRNA-inhibitor control (negative control A probe), or (3) saline (PBS, sham group). Mice received four i.p. doses every three days at 9 mg/kg (miRNA-inhibitor or miRNA-inhibitor control) in 200 µL saline solution (PBS, RNase free). Twenty-four hours after the last injection (or 10 days after the first one), mice were euthanized by i.p. ketamine (75 mg/kg)–xylazine (10 mg/kg) injection. Blood was collected via cardiac puncture. The spleen and PLN were harvested and processed. Pancreases were harvested, snap-frozen in an isopentane/cold acetone bath and stored at −80 °C until use.

### 4.10. Insulitis Score

The degree of islet infiltration by leukocytes (insulitis) was determined in the pancreas of six mice per group at the end of the study. Briefly, non-overlapping cryosections of 6 µm were obtained, placed on a slide, and stained with hematoxylin and eosin. To score insulitis, a minimum of 40 islets per animal were analyzed under a light microscope, as previously described [77]: 0, intact islets/no insulitis; 1, peri-islet infiltrates; 2, <25% islet infiltration; 3, 25–75% islet infiltration; and 4, >75% islet infiltration or complete islet destruction. A double-blind analysis was performed by independent observers.

### 4.11. Flow Cytometry

To determine changes in the percentage of immune cell subpopulations induced by miRNA blockade, the spleen and PLN of all mice were immunophenotyped by flow cytometry.

#### 4.11.1. Leukocyte Isolation from Spleen and PLN

Splenocyte and leukocyte suspensions were obtained by the mechanical disruption of the spleen and PLN, respectively, and washed twice with RPMI-1640 (Biowest, Nuaille, France) + 10% fetal bovine serum (Gibco, Invitrogen, Carlsbad, CA, USA) (R-10) in order to collect all the cells. The tissue remains were allowed to precipitate for 2 min, and the supernatant was obtained, which was then centrifuged at 400× *g* for 5 min at room temperature. Afterward, in the case of the splenocyte suspension, erythrocytes were lysed with 5 mL of hemolysis solution [500 mL deionized H_2_O (Milli-Q Direct, Merck Millipore, Burlington, MA, USA) plus 1.0297 g Trizma Hydrochloride (Sigma-Aldrich, Saint Louis, MO, USA) and 3.735 g NH_4_Cl (Probus, Badalona, Spain)] for 5 min at room temperature, which was next blocked by adding 5 mL of R-10. Cells were centrifuged again at 400× *g* for 5 min at room temperature and further washed with another 5 mL of R-10. Finally, leukocyte suspensions from both spleen and PLN were resuspended in 1–3 mL and 200 µL of PBS + 2% fetal bovine serum, respectively.

#### 4.11.2. Viability and Cell Counting

To assess cell viability and counting, 10 µL of cells were incubated with 2 µL of 7-aminoactinomicina D (7-AAD, BD Biosciences) in 48 µL of PBS for 15 min at room temperature and protected from light. Then, 10 µL of Perfect Count Microspheres (Cytognos SL, Salamanca, Spain) were added to perform cell counting. Cells were acquired by FACSCanto II flow cytometer (BD Biosciences) using the FACSDiva software (BD Biosciences).

#### 4.11.3. Immunophenotype

Next, the percentages of T and B lymphocytes and DCs were assessed by flow cytometry. For phenotype labeling, 0.5 × 10^6^ cells per panel were used; one of them was designed for the study of different T lymphocyte subpopulations (panel 1), and the other was designed for the detection of conventional DCs (panel 2). For the T lymphocyte panel, surface staining was carried out with CD3 PE, CD4 APC, CD8 V500, CD44 BV786, CD62L APC-Cy7, PD-1 PE-Cy7, and CD25 PerCP-Cy5.5, and intracellular staining was carried out with FoxP3 FITC. For the DC panel, surface staining was carried out with CD3 PE, CD19 V450, CD11c PE-Cy7, and MHC-II APC. Further information on antibodies used can be found in Table 4.

First, surface staining was performed. Cells were incubated with the appropriate monoclonal antibodies for 20 min at 4 °C in the dark. If no intracellular staining was needed, cells were then washed with FACSFlow™ Sheath Fluid (ThermoFisher Scientific) and resuspended in the same solution for their acquisition or, for the DC panel, incubated with 4 µL of 7-AAD in 100 µL of PBS for 15 min at room temperature and protected from light. In the case of the intracellular staining with FoxP3, cells were washed with cold FACSFlow™ Sheath Fluid (ThermoFisher Scientific) at 400× *g* for 5 min and fixed with 1 mL of Fixation/Permeabilization solution (1 Fix/Perm concentrate: 3 diluent solution, ThermoFisher Scientific) for 45 min at 4 °C in the dark. After the incubation, cells were washed twice with 2 mL of 1X Permeabilization buffer (400× *g*, 5 min, at room temperature), and the supernatant was discarded. Then, fixed and permeabilized cells were stained with FoxP3 FITC for 40 min at 4 °C in the dark. After that, cells were washed twice with 2 mL 1X Permeabilization buffer at 400× *g* for 5 min and suspended in 200 µL FACSFlow™ Sheath Fluid (ThermoFisher Scientific). At least 100.000 leukocyte events per sample were acquired using FACSCanto II and LSR Fortessa flow cytometers (BD Biosciences). Necrotic and apoptotic cells were excluded from the analysis based on their forward scatter-A/side scatter-A properties and doublets were excluded by forward scatter-A/forward scatter-H. Fluorescence minus one controls were used to assess PD-1, CD25, and FoxP3 staining positivity. Data were analyzed using FlowJo software (Tree Star Inc., Ashland, OR, USA).

### 4.12. Statistical Analysis

Data are presented as mean ± SD or SEM or as percentages, where appropriate. The distribution of continuous variables was tested for normality by the Kolmogorov–Smirnov test. For non-normally distributed variables, comparisons between two groups were performed by a non-parametric Mann–Whitney test, and comparisons between three or more groups were performed by Kruskal–Wallis with Dunn’s post hoc test. Correlations between variables were tested by using a two-tailed Spearman’s test. Weight levels of mice throughout the study were compared using two-way ANOVA with Tukey’s multiple comparisons test. Multivariate statistical analysis was performed using principal component analysis. A Fisher’s exact test meta-analysis method with Benjamini–Hochberg’s FDR correction was used to calculate the significant targeted biological processes and KEGG pathways. The complete linkage clustering method was used for the hierarchical clustering of pathways and miRNAs. Squared Euclidian distances were determined as distance measures, absolute *p*-values were used in all calculations, and the significance levels of the interaction were taken into consideration. For all tests, a two-tailed *p*-value of ≤0.05 was considered statistically significant. Levels of significance are indicated as: *, *p* ≤ 0.05; **, *p* ≤ 0.01; ***, and *p* ≤ 0.001. Analyses were performed using the programs GraphPad Prism 9 (GraphPad Software Inc, San Diego, CA, USA) and R v4.1.0.

### 4.13. Ethics

#### 4.13.1. Human Samples

All the experiments were carried out in strict accordance with the principles outlined in the Declaration of Helsinki for human research and after the approval of the Committee on the Ethics of Research of the Germans Trias i Pujol University Hospital and Parc Taulí University Hospital.

#### 4.13.2. Mice

This study was carried out in strict accordance with the recommendations in the Guide for the Care and Use of Laboratory Animals of the Generalitat de Catalunya. The procedures carried out with animal models were authorized by the Animal Experimentation Ethics Committee of the CMCiB and IGTP and by the Generalitat de Catalunya, and they followed the principles outlined in the Declaration of Helsinki for animal experimental investigation. All the conducted protocols followed the principles of the 3R, prioritizing the welfare of animals used in research.

## 5. Conclusions

Circulating microRNAs have been found to be altered in people with recent-onset T1D, although there are no studies concerning PR (honeymoon). Here, we identify a unique plasma microRNA signature during this phase, providing new microRNA candidate biomarkers for the monitoring of PR in pediatric patients with T1D, which may be used for patient stratification and applied in clinical research. Furthermore, this study demonstrates for the first time that miR-30d-5p—an upregulated miRNA during PR—can modulate various immunoregulatory parameters such as Treg levels or PD-1 expression in the NOD mouse model, shedding light on the mechanisms underlying this phase.

## Figures and Tables

**Figure 1 ncrna-09-00017-f001:**
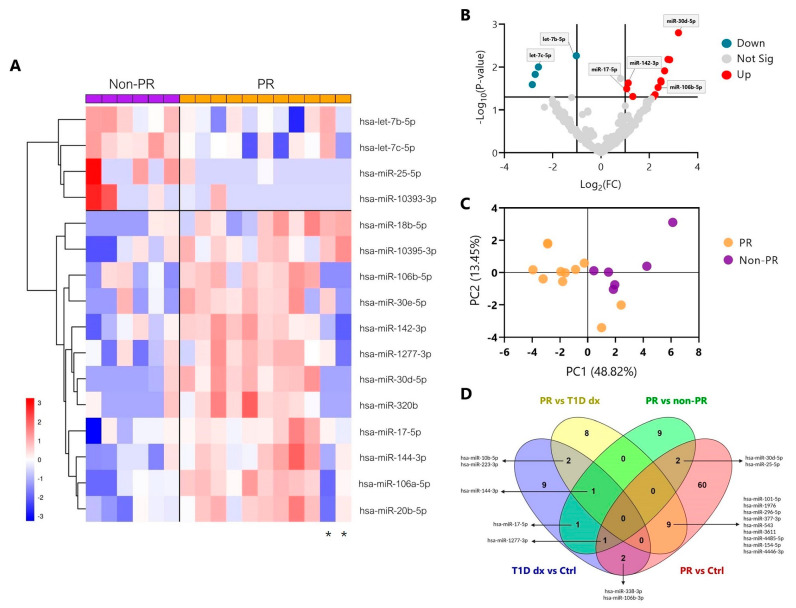
**Distinct miRNA signature in plasma during the partial remission phase of T1D in pediatric patients.** (**A**) Hierarchical clustering heatmap showing the 16 miRNAs with significantly different expression levels (DEMs) in plasma among pediatric patients with T1D in remission (PR, orange) and without remission (non-PR, violet) from the discovery cohort. Each column represents individual samples, and each row represents an individual miRNA. Upregulated miRNAs are shown in red, and downregulated miRNAs are shown in blue. An asterisk below two patients in PR marks their different miRNA expression patterns compared to other remitters. (**B**) Volcano plot showing changes in miRNA levels between pediatric patients with T1D in remission and without remission. Lines indicate log(FC) (*x*−axis) and *p*-value (*y*−axis) cut-offs. Blue and red dots indicate significantly downregulated and upregulated miRNAs during PR, respectively, and grey dots indicate non-significantly different expression levels of miRNAs. (**A**,**B**) Log(FC) > 1 for upregulated DEMs, and log(FC) < 1 for downregulated DEMs. *p*-value < 0.05 by moderated *t*-test; *n* = 6–11. (**C**) Principal component analysis plot of the 16 DEMs. Each dot represents a sample. Orange: remitter patients (*n* = 11); violet: non-remitter patients (*n* = 6). (**D**) Venn diagrams showing overlapping DEMs with validated target genes between different comparisons (PR vs. non-PR; PR vs. controls (Ctrl); PR vs. T1D diagnosis (dx), T1D diagnosis vs. controls).

**Figure 2 ncrna-09-00017-f002:**
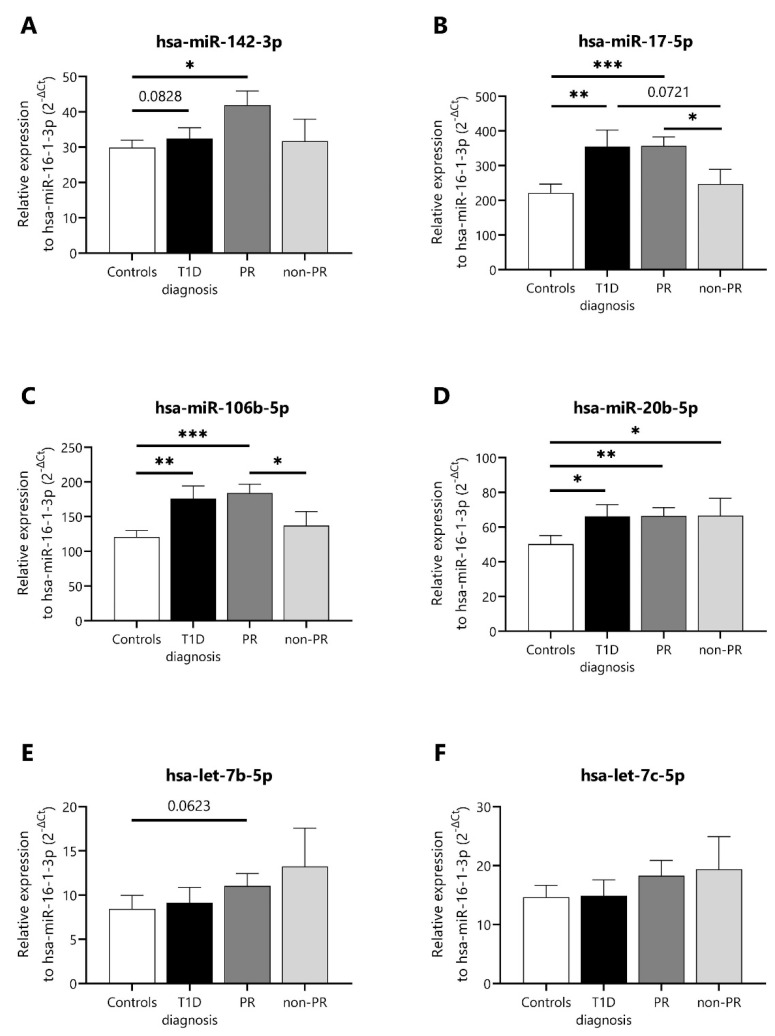
**Differentially expressed miRNAs in plasma of T1D patients at different stages and non-diabetic control subjects.** (**A**–**F**) Single-assay RT-qPCR validation of some differentially expressed miRNAs in plasma samples of controls (*n* = 15), newly diagnosed patients with T1D (*n* = 8), remitter patients (PR, *n* = 10), and non-remitter patients (non-PR, *n* = 9) from the validation cohort. The miRNA expression signal was normalized to hsa-miR-16-1-3p expression. Values are expressed as 2^-∆Ct^. Data are presented as mean ± SEM. * *p* ≤ 0.05, ** *p* ≤ 0.01, *** *p* ≤ 0.001, Kruskal–Wallis with Dunn’s post hoc test.

**Figure 3 ncrna-09-00017-f003:**
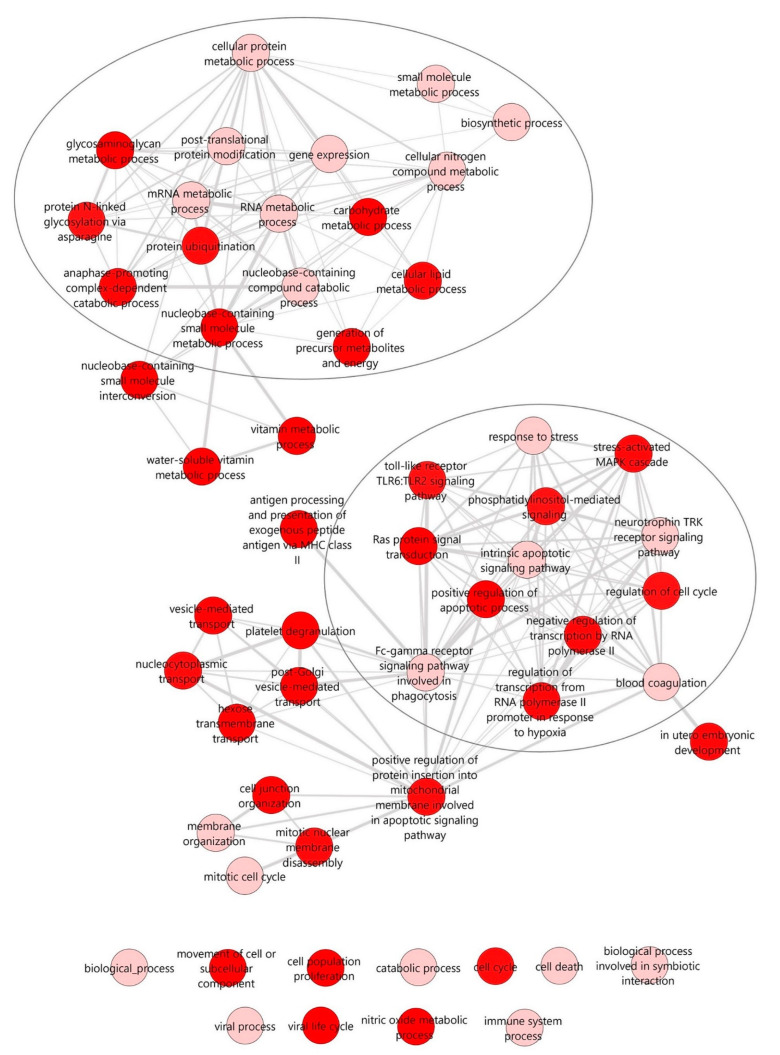
**Interactive graph of the enriched GO terms for the selected miRNAs.** The interactive graph was generated using the web tool REVIGO (http://revigo.irb.hr/, accessed on 21 January 2022), which plots the remaining relevant GO terms (after reducing redundancy) for the 14 DEMs with validated target genes that were retrieved from the DIANA-miRPath v3 (http://www.microrna.gr/miRPathv3, accessed on 21 January 2022) web server using the TarBase v7 database. Highly similar GO terms are linked by edges in the graph, where the line width indicates the degree of similarity. The bubble color indicates the provided *p*-value for the FDR (q-value). Darker red color indicates statistically more significant GO terms. Two main clusters of biological processes were identified: one comprising different signaling pathways related to the immune response, apoptosis, or stress (**upper** circle) and the other comprising metabolic processes (**lower** circle).

**Figure 4 ncrna-09-00017-f004:**
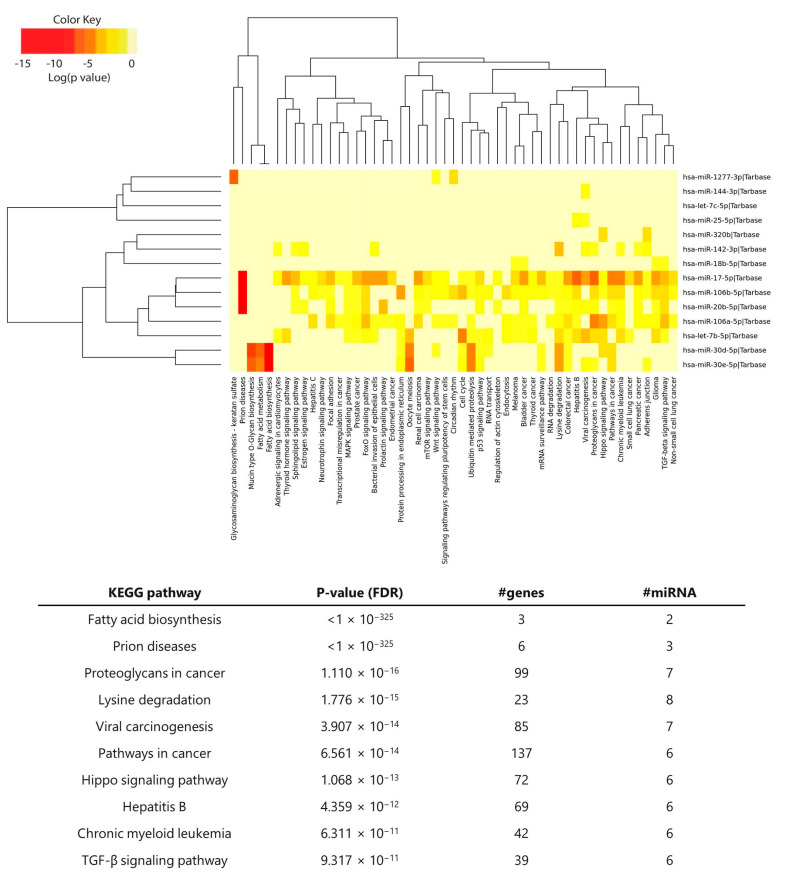
**KEGG pathway enrichment analysis of the selected miRNAs.** The heatmap shows the significantly enriched pathways (*q*-value < 0.05) of the selected miRNAs retrieved from the KEGG database in the DIANA-miRPath v3 web server (http://www.microrna.gr/miRPathv3, accessed on 13 September 2022). Hierarchical clustering of miRNAs was performed based on their similar pathway targeting patterns, and pathways were clustered together by related miRNAs. The color code represents the log (*p*-value (FDR)), with the most significant miRNA-pathway interactions in red. The table shows the top 10 most significant KEGG pathways and the related *p*-values (FDR), the number of genes involved in the pathway, and the number of miRNAs targeting the indicated pathways.

**Figure 5 ncrna-09-00017-f005:**
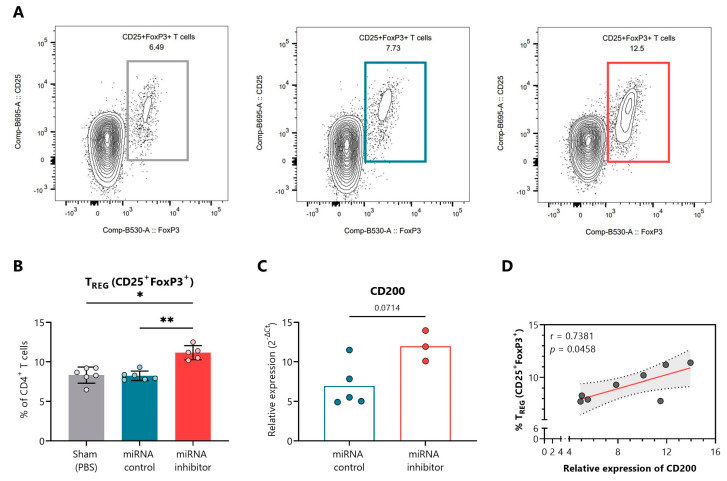
**In vivo miR-30d-5p inhibition promotes Treg expansion by upregulating *CD200* in pancreatic lymph nodes.** (**A**) Representative FACS plots indicating the percentage of CD4^+^CD25^+^FoxP3^+^ Tregs in the PLN of the sham (**left**, grey), control inhibitor (**middle**, blue), and miRNA inhibitor (**right**, red) treatment groups. (**B**) Percentage of total CD4^+^CD25^+^FoxP3^+^ Tregs in the PLN of sham mice (*n* = 6, **left**, grey), mice treated with a control inhibitor (*n* = 6, **middle**, blue), or mice treated with the miR-30d-5p inhibitor (*n* = 5, **right**, red). Data are presented as mean ± SD. * *p* ≤ 0.05, ** *p* ≤ 0.01, Kruskal–Wallis with Dunn’s post hoc test. (**C**) Levels of *CD200* mRNA (a validated target of miR-30d-5p) on the remaining PLN cells of mice treated with a control inhibitor (*n* = 5, **left**, blue) or the miR-30d-5p inhibitor (*n* = 3, **right**, red). Gene expression signal was normalized to *GAPDH* expression. Values are expressed as 2^−∆Ct^. Data are presented as mean ± SD. *p*-value was derived by the Mann–Whitney test. (**D**) Correlation between levels of *CD200* mRNA and the percentage of total Tregs (CD4^+^CD25^+^FoxP3^+^). A significant positive correlation was found between both parameters (Spearman’s r = 0.7381, *p* = 0.0458, Spearman’s correlation analysis).

**Figure 6 ncrna-09-00017-f006:**
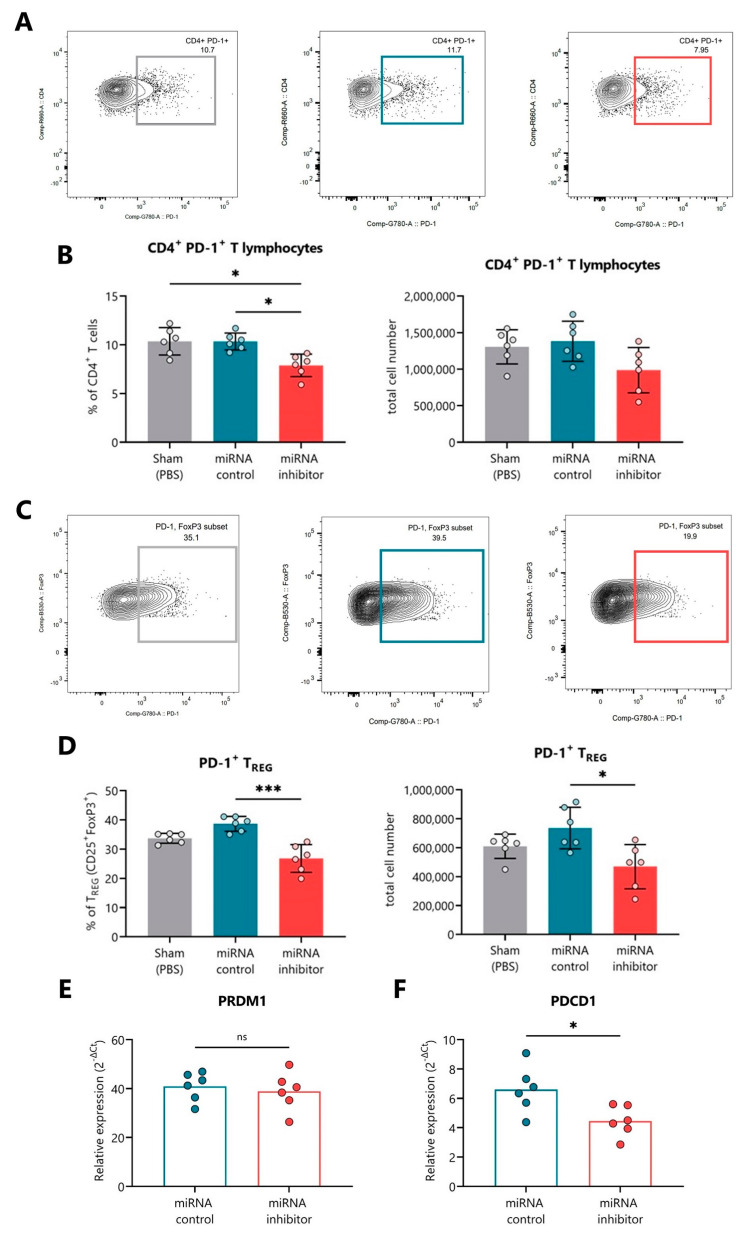
**In vivo miR-30d-5p inhibition is related to decreased levels of PD-1 in the spleen.** (**A**,**C**) Representative FACS plots indicating the percentage of (**A**) CD4^+^PD-1^+^ T lymphocytes or (**C**) PD-1^+^ Tregs in the spleen of the sham (**left**, grey), control inhibitor (**middle**, blue), and miRNA inhibitor (**right**, red) treatment groups. (**B**,**D**) Percentage (**left**) and total cell numbers (**right**) of (**B**) CD4^+^PD-1^+^ T lymphocytes or (**D**) PD-1^+^ Tregs in the spleen of sham mice (*n* = 6, **left**, grey), mice treated with a control inhibitor (*n* = 6, **middle**, blue), or mice treated with the miR-30d-5p inhibitor (*n* = 6, **right**, red). Data are presented as mean ± SD. * *p* ≤ 0.05, *** *p* ≤ 0.001 Kruskal–Wallis with Dunn’s post-hoc test. (**E**,**F**) Levels of (**E**) *PRDM1* (a validated target of miR-30d-5p and a repressor of *PDCD1*) and (**F**) *PDCD1* mRNAs in the splenocytes of mice treated with a control inhibitor (*n* = 6, **left**, blue) or the miR-30d-5p inhibitor (*n* = 6, **right**, red). Gene expression signal was normalized to *GAPDH* expression. Values are expressed as 2^−∆Ct^. Data are presented as mean ± SD. * *p* ≤ 0.05, Mann–Whitney test.

**Figure 7 ncrna-09-00017-f007:**
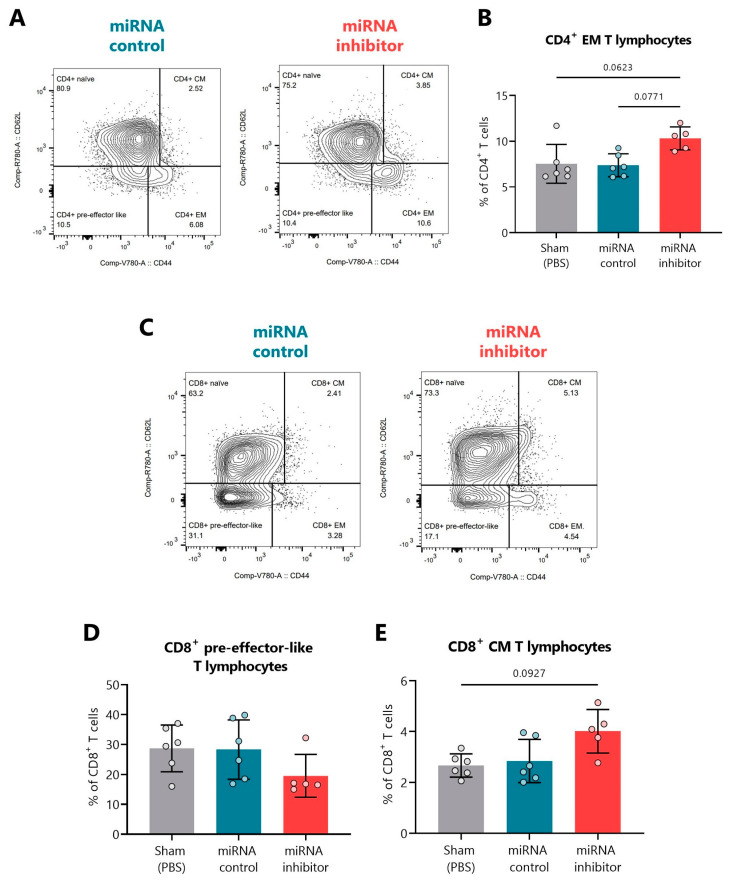
**In vivo miR-30d-5p inhibition resulted in T cell subset changes in the pancreatic lymph nodes.** (**A**) Representative FACS plots indicating the percentage of CD4^+^ naïve, EM, CM, and pre-effector-like T cells in PLN of the control inhibitor (**left**, blue) and miRNA inhibitor (**right**, red) treatment groups. (**B**) Percentage of CD4^+^ EM T cells in PLN of sham mice (*n* = 6, **left**, grey), mice treated with a control inhibitor (*n* = 6, middle, blue), or mice treated with the miR-30d-5p inhibitor (*n* = 5, **right**, red). (**C**) Representative FACS plots indicating the percentage of CD8^+^ naïve, EM, CM, and pre-effector-like T cells in PLN of the control inhibitor (**left**, blue) and miRNA inhibitor (**right**, red) treatment groups. (**D**,**E**) Percentages of (**D**) CD8^+^ pre-effector-like T cells or (**E**) CD8^+^ CM T cells in PLN of sham mice (*n* = 6, **left**, grey), mice treated with a control inhibitor (*n* = 6, **middle**, blue) or mice treated with the miR-30d-5p inhibitor (*n* = 5, **right**, red). Data are presented as mean ± SD. *p*-values were determined by Kruskal–Wallis with Dunn’s post hoc test.

**Figure 8 ncrna-09-00017-f008:**
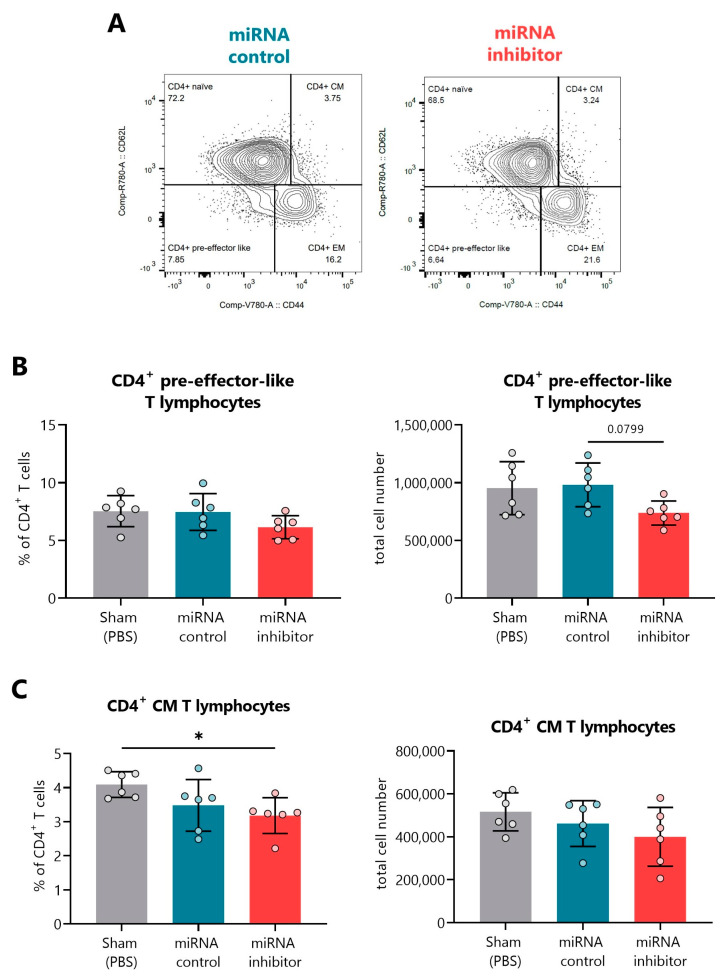
**In vivo miR-30d-5p inhibition decreases both CD4^+^ pre-effector-like and CM T cells in the spleen.** (**A**) Representative FACS plots indicating the percentage of CD4^+^ naïve, EM, CM, and pre-effector-like T cells in the spleen of the control inhibitor (**left**, blue) and miRNA inhibitor (**right**, red) treatment groups. (**B**,**C**) Percentage (**left**) and total cell numbers (**right**) of (**B**) CD4^+^ pre-effector-like T cells and (**C**) CD4^+^ CM T cells in the spleen of sham mice (*n* = 6, **left**, grey), mice treated with a control inhibitor (*n* = 6, **middle**, blue), or mice treated with the miR-30d-5p inhibitor (*n* = 6, **right**, red). Data are presented as mean ± SD. * *p* ≤ 0.05, Kruskal–Wallis with Dunn’s post hoc test.

**Figure 9 ncrna-09-00017-f009:**
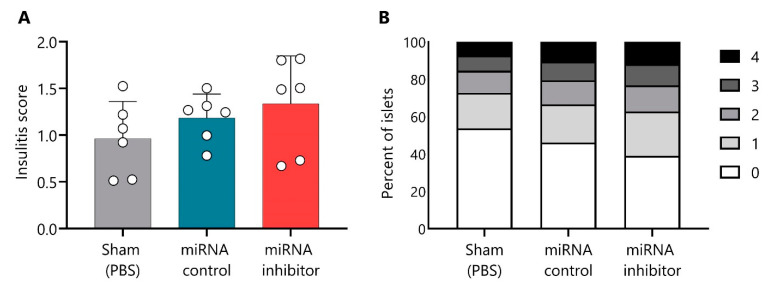
**Slight increase in insulitis after a short treatment with miR-30d-5p inhibitor.** (**A**) Insulitis score from prediabetic NOD mice at the end of the short treatment (10 weeks and 3 days) with PBS (sham, *n* = 6, **left**, grey), control inhibitor (*n* = 6, **middle**, blue), or the miR-30d-5p inhibitor (*n* = 6, **right**, red). Data are presented as mean ± SD. (**B**) Percentage of islets in each of the infiltration categories: white = 0, no insulitis; light grey = 1, peri-insular; medium grey = 2, mild insulitis (<25% of the infiltrated islet); dark grey = 3, 25–75% of the islet infiltrated; black = 4, >75% islet infiltration. Significant differences were not found between groups.

**Table 1 ncrna-09-00017-t001:** Clinical and metabolic data of patients with T1D from both the discovery and validation cohorts.

	Discovery Cohort				Validation Cohort			
	Ctrl	T1D dx	PR	Non-PR	Ctrl	T1D dx	PR	Non-PR
**N (no. girls)**	17 (10)	17 (10)	11 (6)	6 (4)	15 (9)	8 (4)	10 (5)	9 (7)
**Age at diagnosis (years)**	8.8 ± 3.4	8.7 ± 3.6	9.1 ± 4.3	9 ± 2.8	9.7 ± 3.6	11.6 ± 2.7	11.8 ± 2.9	7.3 ± 3.7
**BMI (kg/m^2^)**	18.3 ± 4.3	16.8 ± 2.5	17.7 ± 3	17.2 ± 2.1	19 ± 4.2	19.4 ± 4.3	20.2 ± 4.6	16.9 ± 1.7
**HbA1c (%)**	ND	11.4 ± 2.4	6.9 ± 0.6	8.1 ± 0.7	ND	12.6 ± 2.5	6.4 ± 0.6	8.3 ± 0.6
**Insulin dose (U/kg/day)**	ND	0.7 ± 0.2	0.4 ± 0.1	0.9 ± 0.1	ND	0.77 ± 0.2	0.46 ± 0.2	0.87 ± 0.2
**IDAA1c**	ND	14.3 ± 3	8.4 ± 0.5	11.5 ± 1	ND	15.7 ± 3.1	8.2 ± 0.79	11.8 ± 1.4
**Basal C-peptide (ng/mL)**	1.3 ± 0.4	0.3 ± 0.2	0.7 ± 0.5	0.3 ± 0.1	1.97 ± 0.4	0.44 ± 0.3	1 ± 0.57	0.2 ± 0.1

Data presented as mean ± SD. Abbreviations: BMI, body mass index; Ctrl, controls; HbA1c, glycated hemoglobin; IDAA1c, insulin dose-adjusted HbA1c; ND, not determined; non-PR, non-partial remission; PR, partial remission; T1D dx, type 1 diabetes diagnosis.

**Table 2 ncrna-09-00017-t002:** Differentially expressed circulating miRNAs with validated target genes during the partial remission phase.

miRNAs	PR vs. Ctrl (FC)	*p*-Value	PR vs. T1D dx (FC)	*p*-Value	PR vs. Non-PR (FC)	*p*-Value
hsa-miR-30d-5p	1.777	0.019	-	-	3.208	0.001
hsa-let-7b-5p	-	-	-	-	−1.019	0.005
hsa-miR-106a-5p	-	-	-	-	2.771	0.006
hsa-miR-20b-5p	-	-	-	-	2.842	0.006
hsa-let-7c-5p	-	-	-	-	−2.587	0.009
hsa-miR-25-5p	−1.921	0.019	-	-	−2.722	0.014
hsa-miR-320b	-	-	-	-	2.477	0.020
hsa-miR-30e-5p	-	-	-	-	2.485	0.022
hsa-miR-142-3p	-	-	-	-	1.125	0.023
hsa-miR-106b-5p	-	-	-	-	2.361	0.030
hsa-miR-144-3p	-	-	−1.027	0.006	1.064	0.032
hsa-miR-18b-5p	-	-	-	-	2.235	0.043
hsa-miR-17-5p	-	-	-	-	1.319	0.048
hsa-miR-1277-3p	2.126	0.008	-	-	2.162	0.049
hsa-miR-101-5p	−1.935	0.015	−2.910	0.002	-	-
hsa-miR-4485-3p	-	-	−2.890	0.005	-	-
hsa-miR-10b-5p	-	-	2.697	0.005	-	-
hsa-miR-1976	−1.568	0.037	−2.433	0.008	-	-
hsa-miR-296-5p	−1.830	0.018	−2.444	0.009	-	-
hsa-miR-377-3p	−2.028	0.016	−2.558	0.013	-	-
hsa-miR-543	−1.838	0.023	−2.311	0.018	-	-
hsa-miR-24-3p	-	-	2.069	0.019	-	-
hsa-miR-3611	−2.293	0.002	−2.108	0.019	-	-
hsa-miR-4485-5p	−-2.883	0.005	−2.846	0.020	-	-
hsa-miR-154-5p	−2.556	0.001	−2.256	0.020	-	-
hsa-miR-223-3p	-	-	1.311	0.021	-	-
hsa-miR-324-3p	-	-	−2.244	0.022	-	-
hsa-miR-1-3p	-	-	−2.263	0.029	-	-
hsa-miR-4449	-	-	−1.951	0.037	-	-
hsa-miR-365a-3p	-	-	2.059	0.039	-	-
hsa-miR-365b-3p	-	-	2.059	0.039	-	-
hsa-miR-4446-3p	−2.194	0.009	−2.025	0.043	-	-
hsa-miR-132-3p	-	-	−1.943	0.045	-	-

Data analyzed by moderated *t*-test. Abbreviations: Ctrl, controls; FC; fold change; non-PR, non-partial remission; *PR*, partial remission; *T1D dx*, type 1 diabetes diagnosis.

**Table 3 ncrna-09-00017-t003:** TaqMan Assays used for analyzing targeted miRNA expression in plasma samples.

miRNA	Assay ID
hsa-miR-142-3p	477910_mir
hsa-miR-20b-5p	477804_mir
hsa-miR-17-5p	478447_mir
hsa-let-7b-5p	478576_mir
hsa-let-7c-5p	478577_mir
hsa-miR-106b-5p	478412_mir
ath-miR159a	478411_mir
hsa-miR-16-1-3p	478727_mir

**Table 4 ncrna-09-00017-t004:** List of antibodies used for spleen and PLN immunophenotyping.

	Target	Fluorophore	Species/Isotype	Clone	Use/100 µL	Company
PANEL 1	CD3	PE	Hamster IgG	500A2	1	BD Biosciences
	CD4	APC	Rat IgG2a, κ	RM4-5	0.5	BD Biosciences
	CD8	V500	Rat IgG2a, κ	53-6.7	0.4	BD Biosciences
	CD44	BV786	Rat IgG2b, κ	IM7	0.2	BD Biosciences
	CD62L	APC-Cy7	Rat IgG2a, κ	MEL-14	0.4	BioLegend
	PD-1	PE-Cy7	Rat IgG2a, κ	29F.1A12	2	BioLegend
	CD25	PerCP-Cy5.5	Rat/IgG1, λ	PC61.5	1	eBioscience
	FOXP3	FITC	Rat IgG2a, κ	FJK-16s	1.5	eBioscience
PANEL 2	CD3	PE	Hamster IgG	500A2	1	BD Biosciences
	CD19	V450	Rat IgG2a, κ	1D3	1	BD Biosciences
	CD11c	PE-Cy7	Hamster IgG1, λ2	HL3	1	BD Biosciences
	MHC-II	APC	Rat IgG2b, κ	AMS-32.1	1	eBioscience

## Data Availability

All data produced or analyzed in this study are contained in this published article and its Appendix A. The datasets generated and/or analyzed during the current study are available in the European Nucleotide Archive at EMBL-EBI repository under accession number PRJEB58187 (https://www.ebi.ac.uk/ena/browser/view/PRJEB58187, accessed 22 December 2022) and/or are accessible by the corresponding authors upon reasonable request.

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
