# Peer review of "Immunoregulatory Biomarkers of the Remission Phase in Type 1 Diabetes: miR-30d-5p Modulates PD-1 Expression and Regulatory T Cell Expansion"

_ncrna, 2023, doi:10.3390/ncrna9020017_

Round 1

Reviewer 1 Report

In this manuscript, the authors analyze miRNA signatures in the partial remission (PR) phase of patients with Type 1 Diabetes. This work is original and novel, as until now the miRNA signatures during the so called honeymoon phase in T1D have not been studied. The methodological approaches are appropriate, the sample size, although somewhat modest, is sufficient to draw statistically significant results, and the analysis is correct. The results are very well explained, and the conclusions of the study are coherent with the results.

Only some minor issues remain:

-          Page 3, lines 124-126 and Figure 1: “Nevertheless, two remitter patients are clustered within the non-remitter group, which coincides with the two of the heatmap that showed a distinct profile in comparison to the other PR patients, one of them presenting also celiac disease”. To ease the readers’ interpretation of this result, it would be needed to highlight which these two patients are in Figure 1.

-          Which are the inclusion/exclusion criteria for non-diabetic children? This should be shown in the Methods section.

-          Page 5, line 163: “forty-seven”. Is this figure correct?

-          Page 5, line 168: miR-142-3p is not significantly upregulated in PR versus non-PR (as shown in Figure 2A), please correct.

-          Was miR-30d-5p not analyzed in the validation cohort?

-          Page 11, lines 241-242: “viability of splenocytes and PLN cells was optimal in the three groups (Figure S2).”. This should be toned down, as the figure does show statistically significant differences, although the reviewer agrees that the degree of the decrease is not high.

-          Page 12, line 273: the word “impairs” should be replaced by a term that more accurately reflects the results shown by the authors, as the data does not show a total impairment of PD1 expression, but a reduction in the expression levels.

-          Page 20, line 442: should read “comparison” instead of “compassion”.

-          Page 25, lines 681, 688 and 689: the acronym “R-10” should be explained.

Reviewer 2 Report

The work of Gomez-Munoz and colleagues is very well written and highlights a very important aspect of T1D, namely its heterogeneity in clinical and molecular terms, which is reflected in the search for potential circulating biomarkers in the pivotal period of partial remission, which, however, occurs only in some patients but not in others.

However, there are some small points to be clarified:

- Regarding the clinic, the paper given as reference 36 for choosing IDAA1c<9 to distinguish PR from non-PR is accurate, but why 8 months post-diagnosis?

- Since several comparisons were made in the differential analysis, it would be appropriate to include the number of DE miRNAs between one group and another, since from the table alone it is difficult to trace the 16 DEs between PRs and non-PRs and follow those who are DEs in the other comparisons.

-The authors should include the method of plasma isolation as it is a very important pre-analytical variable that strongly influences the subsequent expression analisis of miRNAs.

- Why was hemolysis evaluated only in the validation but not in the discovery cohort?

- Since 6 sequencing runs were performed, were repeated technical replicates included between runs? If yes, give details; if no, cite it as a limitation.

- For PCR normalization, is the normalizer used miR-16-5p (as shown in materials and methods) or miR-16-1-3p (as shown in figure s6 and figure legend of figure 2?). Also, since the Ct of the normalizer is 28-29 (a bit high) it would be better to also use another normalizer with higher expression and check their reliability.

- I cannot catch the importance of miR-30d-5p from the way the paper is written. It almost seems that other miRNAs (miR-17 family) are more important. It would be good to highlight its importance better. For example, since a pathways analysis was done, maybe it is the one most involved in something particularly interesting related to T1D?
